# Gene Expression and Isoform Identification of PacBio Full-Length cDNA Sequences for Berberine Biosynthesis in *Berberis koreana*

**DOI:** 10.3390/plants10071314

**Published:** 2021-06-28

**Authors:** Neha Samir Roy, Ik-Young Choi, Taeyoung Um, Mi Jin Jeon, Bo-Yun Kim, Young-Dong Kim, Ju-Kyung Yu, Soonok Kim, Nam-Soo Kim

**Affiliations:** 1Agriculture and Life Sciences Research Institute, Kangwon National University, Chuncheon 24341, Korea; neha_roy@kangwon.ac.kr (N.S.R.); choii@kangwon.ac.kr (I.-Y.C.); tyoungum@kangwon.ac.kr (T.U.); 2Microorganism Resources Division, National Institute of Biological Resources, Incheon 22689, Korea; mj428star@korea.kr; 3Plant Resources Division, National Institute of Biological Resources, Incheon 22689, Korea; bykim416@korea.kr; 4Department of Life Science, Multidisciplinary Genome Institute, Hallym University, Chuncheon 24252, Korea; ydkim@hallym.ac.kr; 5Syngenta Crop Protection LLC, 9 Davis Drive, Research Triangle Park, NC 27709, USA; yjk0830@hotmail.com; 6Department of Molecular Bioscience, Kangwon National University, Chuncheon 24341, Korea

**Keywords:** *Berberis koreana*, benzylisoquinoline alkaloids (BIAs), berberine, isoforms, PacBio Iso-Seq

## Abstract

*Berberis koreana* is a medicinal plant containing berberine, which is a bioactive compound of the benzylisoquinoline alkaloid (BIA) class. BIA is widely used in the food and drug industry for its health benefits. To investigate the berberine biosynthesis pathway, gene expression analysis was performed in leaves, flowers, and fruits at different stages of growth. This was followed by full-length cDNA sequencing analysis using the PacBio sequencer platform to determine the number of isoforms of those expressed genes. We identified 23,246 full-length unigenes, among which 8479 had more than one isoform. The number of isoforms ranged between two to thirty-one among all genes. Complete isoform analysis was carried out on the unigenes encoding BIA synthesis. Thirteen of the sixteen genes encoding enzymes for berberine synthesis were present in more than one copy. This demonstrates that gene duplication and translation into isoforms may contribute to the functional specificity of the duplicated genes and isoforms in plant alkaloid synthesis. Our study also demonstrated the streamlining of berberine biosynthesis via the absence of genes for enzymes of other BIAs, but the presence of all the genes for berberine biosynthesize in *B. koreana*. In addition to genes encoding enzymes for the berberine biosynthesis pathway, the genes encoding enzymes for other BIAs were not present in our dataset except for those encoding corytuberine synthase (CTS) and berbamunine synthase (BS). Therefore, this explains how *B. koreana* produces berberine by blocking the pathways leading to other BIAs, effectively only allowing the pathway to lead to berberine synthesis.

## 1. Introduction

The genus *Berberis* (Berberidaceae), also known as barberry, is a large genus containing approximately 500 species that are distributed throughout temperate and subtropical regions of the world [1,2]. The barberry plants are spiny, deciduous, evergreen shrub- or small tree-bearing berry fruits that are long and ripen red or dark blue [3]. Many *Berberis* species are known to have diverse phytochemicals, such as various alkaloids, terpenoids, flavonoids, sterols, anthocyanins, lignans, lipids, and carotenoids, so they have been used in traditional medicine in various parts of the world since ancient times [1]. Berberine, the main compound in *Berberis* species, is an organic heteropentacyclic compound in the protoberberine group of benzylisoquinoline alkaloids (BIAs) (https://pubchem.ncbi.nlm.nih.gov/compound/2353 (accessed on 3 May 2021)). BIAs include a diverse class of nitrogen-containing plant secondary metabolites, with approximately 2500 molecules, such as morphine, codeine, sanguinarine, and Papaverine [4,5]. BIAs are present in only restricted plant families, including *Berberidaceae*, *Papaveraceae*, and *Ranunculaceae* in Ranunculales; *Fabaceae* in Fabales; and *Magnoliaceae* in Magnoliales [6,7].

*Berberis koreana* (*B. koreana*) is an endemic medicinal plant in Korea [8]. It can grow approximately 1.5 m in height and bears berries that are purple to red. Biological and pharmaceutical activities such as anticancer and antioxidant activities [9], neuroprotective effects against ischaemic damage [10], and anti-inflammatory [11] and cytotoxic effects [12] were reported in extracts from *B. koreana*. The main constituent of the *B. koreana* extract is berberine [12]. The pharmacological and biological effects of berberine include antibacterial, antioxidant, anti-inflammatory, anti-convulsion, sedative, anti-cholinergic, cholagogic, hepatoprotective, and anticancer effects [2,13].

The biosynthetic pathways for diverse arrays of BIAs have been established in several plants. Farrow et al. (2012) [4] reported metabolic frameworks determined by integration of transcripts and metabolic profiles in 18 BIA-producing plant species. Transcriptome analysis was also attempted to isolate functional homologous genes involved in the BIA biosynthesis pathways in BIA-producing 20 plants [5]. The researchers successfully isolated novel catalysts within BIA metabolism so that complicated biochemical pathways were elucidated for several BIAs, including papaverine, morphine, sanguinarine, berberine, and noscapine [5]. The presence of BIAs is known in lotus (*Nelumbo nucifera*), but the biosynthetic pathway and regulation are not clear in lotus. Deng et al. (2018) clearly showed the BIA biosynthetic pathway and regulation of the involved genes by transcriptome analysis in lotus. BIA metabolism and tissue-specific accumulation of BIAs were deeply analyzed by combined analyses of transcriptome sequences by single-molecule real-time (SMRT) sequencing and ultra-performance liquid chromatography-electrospray ionization tandem mass spectrometry (UHPLC-ESC-MS/MS) in *Coptis deltoidea* (Papaveraceae) [14].

Conventional transcriptomic analyses utilize high-throughput short-read RNA-Seq data based on the second-generation sequencing technique, which is quite efficient for the quantification of gene expression profiles [15]. A significant portion of eukaryotic genes belongs to gene families, but short-read RNA-Seq analyses limit the identification of multiple full-length transcripts from gene families. Furthermore, eukaryotic genes split into exons and introns, and multiple exon-intron genes are alternatively spliced. Different duplicate copies in family genes and alternative splicing produce different isoforms in different tissues [16] or during development [17]. The SMRT sequencing technique was developed recently by Pacific Biosciences, and a growing number of transcriptome studies performed using the PacBio Iso-Seq protocol have been reported in different organisms (reviewed in Wang et al. 2019 [18], references therein). Third-generation sequencing systems, such as SMRT by PacBio Iso-Seq or single-molecule sequencing (SMS) by Oxford Nanopore sequencing, are advantageous over short-read-based Illumina transcriptome sequencing for comprehensive genome annotation because these new techniques allow the identification of novel gene/isoforms, gene families, long noncoding RNAs, and fusion transcripts [16,19]. In regard to medicinal plants, Jo et al. [20] analyzed transcriptomes in ginseng (*Panax ginseng*) using PacBio Iso-Seq to provide terpenoid saponin synthesis and plant hormone signaling pathways. Kim et al. [21] also reported the transcriptome of *Zanthoxylum planispinum*, a medicinal herb in Korea, using the SMS technique; in that study, 76 cytochrome P450 superfamily members were classified into nine families, and five different isoforms were identified. SMRT sequencing generated 75,438 full-length transcripts, among which, 64 full-length transcripts encoding enzymes involved in BIA synthesis were identified to elucidate the biochemical pathways of diverse BIAs, including berberine, in *C. deltoidea* [14].

For our research, we conducted experiments to understand gene expression and isoform identification in berberine biosynthesis using different tissue types of *B. koreana*. Differential gene expression (DEG) analysis using Illumina short-read sequencing data showed differences in the gene expression encoding berberine biosynthesis among different tissues (i.e., leaf, flower, and fruit). Furthermore, we conducted full-length cDNA sequencing using the SMRT sequencing technique by PacBio to investigate comprehensive gene expression and isoform copy number in berberine biosynthesis. This article also demonstrated the end-to-end process from sequencing to contig assembly to functional annotation (from wet-lab technique to computational analysis).

## 2. Results

### 2.1. Identification and Expression Analysis of Berberine Biosynthesis Pathway Genes

The BIA biosynthetic pathways start with two L-tyrosine molecules, which are converted to dopamine and 4-hydrophenylacetaldehyde (Figure 1). These two molecules are combined by (*S*)-norcoclaurine synthase (NCS) to become (*S*)-norcoclaurine, an immediate precursor to all BIAs. (*S*)-Norcoclaurine is converted to (*S*)-reticuline via a series of enzymatic reactions [5]. (*S*)-Reticuline is then transformed into various BIAs, including berberine, noscapine, sanguinarine, morphine, etc. We identified all the enzymes crucial for the berberine biosynthesis pathway from L-tyrosine to berberine in the transcriptomes in both Illumina and PacBio Iso-Seq libraries. The isolated berberine biosynthesis enzymes were L-tyrosine aminotransferase (TyrAT), tyrosine/tyramine 3-hydroxylase (3OHase), (*S*)-norcoclaurine synthase (NCS), (*RS*)-norcoclaurine 6-*O*-methyltransferase (6OMT), (*S*)-coclaurine N-methyltransferase (CNMT), (*S*)-N-methylcoclaurine 3′-hydroxylase/*N*-methylcoclaurine 3′-monooxygenase (NMCH), 3′-hydroxy-*N*-methyl-(*S*)-coclaurine 4′-*O*-methyltransferase (4ÓMT), berberine bridge enzyme (BBE), and (*S*)-tetrahydroprotoberberine oxidase (STOX), (*S*)-scoulerine 9-*O*-methyltransferase (SOMT), and canadine synthase (CAS/CYP719A21) (Appendix A). Gene families and isoforms of these enzymes are shown in Table 1.

As expected, the genes for enzymes required in the pathways synthesizing other BIAs were not identified in our dataset. For instance, transcripts of reticuline epimerase (REPI) for codeine and morphine synthesis, cheilanthifoline synthase/CYP719A25 for sanguinarine synthesis, norreticuline 7-*O*-methyltransferase (N7OMT) for papaverine synthesis, and tetrahydroprotoberberine *N*-methyltransferase (TNMT) for noscapine synthesis were absent in our study (Figure 1). Although berbamunine and corytuberine were not precursors for berberine synthesis, and unigenes encoding the enzymes, berbamunine synthase (CYP80A1) and corytuberine synthase (CYP80G2) were present in our dataset.

The genes for TYDC, NCS, SOMT, and NMCH were highly expressed in mature leaves and mature fruit, whereas those of 3OHase, 6OMT, and CNMT showed higher expression in young leaves and mature fruit (Figure 1). Genes encoding CAS showed higher expression in young leaves and young fruit, while genes encoding BBE were upregulated in leaves as compared to flower and mature leaves and fruits as compared to younger tissues and 4′OMT showed no particular expression pattern, i.e., 50% of genes were upregulated, and the rest were downregulated.

### 2.2. Berberis Koreana Transcriptome Sequencing by PacBio

A pooled RNA sample of three leaf tissues was sequenced using the PacBio Sequel platform in two SMRT cells with size ranges of <4 kb and >4 kb. Appendix A shows a summary of the results. Figure 2 shows the steps in the computational pipeline used to obtain unique full-length transcripts in *B. koreana*. In the polymerase read, the total number of bases was over 63 Gbp (31 Gbp in the <4 kb library and 32.5 Gbp in the >4 kb library), which yielded 508,050 reads and 570,661 reads in the <4 kb and >4 kb libraries, respectively. In circular consensus sequence (CCS) analysis, we obtained 900,876 CCS reads, which consisted of 455,028 reads from 890 Mbp in the <4 kb library and 445,848 reads from 1.97 Gbp in >4 kb library, respectively. The mean read lengths were 1957 and 4434 bp in the respective libraries. With the aid of standard Iso-Seq classification and clustering protocol (see material and methods) on the obtained CCSs, we produced 43,635 and 32,996 high-quality (HQ) polished transcripts and 261 and 430 low-quality polished transcripts in <4 kb and >4 kb groups, respectively (Appendix A).

With the COding GENome reconstruction Tool (Cogent v7.0.0, https://github.com/Magdoll/Cogent (accessed on 3 May 2021)), the HQ coding sequences (76,631) were further analyzed to generate 19,902 reads from reconstructed coding contigs of 60.49 Mbp. The number of reads in unassigned sequence was 3344. In UniTransModels, the fake genome comprised reconstructed coding contigs and unassigned sequences. Thus, the number of reads in the fake genome was 23,246, and they varied in length from 100 to 13,544 bp with a mean of 3059 bp. Figure 3 shows the size distribution of the reads, with the most frequent length falling between 1000 and 1999 bp, followed by a 4000–4999 bp range.

Of these 23,246 reads, 14,767 unigenes had no isoform, making up 64.21% of the total transcripts. Of the remaining 8479 unigenes, 4812 (20.92%) produced 2 isoforms, followed by 1680 (7.30%) producing 3 isoforms and 728 (3.17%) producing 4 isoforms (Table 2). For isoform analysis, we analyzed in detail the unigenes involved in berberine biosynthesis. Figure 4 shows (*S*)-norcoclaurine synthase (NCS) transcripts exhibiting ten isoforms. NCS catalyzes the formation of norcoclaurine from dopamine and 4-hydroxyphenylacetaldehyde in the berberine synthesis pathway [5].

### 2.3. Functional Annotation

The *B. koreana* transcripts were functionally annotated and categorized by mapping against NCBI databases. Of the 23,246 unigenes, 20,029 (86.16%) and 16,179 (69.9%) were matched with the Nt and Nr databases, respectively. Of those 20,029 unigenes matched to the Nt database, the most unigenes were matched with those of *Nelumbo nucifera* (3006), followed by *Papaver somniferum* (1554) and *Camellia sinensis* (873) (Appendix A). A similar trend was followed in the Nr database, where the most were annotated to *N. nucifera* (15,082), followed by *P. somniferum* (4412); in contrast, the third and fourth most transcripts were matched to *Vitis vinifera* and *Vitis riparia* (Appendix A).

In functional classification, GO terms were assigned to each of the UniTransModels via the BLAST2GO (bioinformatics platform) program based on annotation of the Nr database. Overall, 16,756 (53.18%) unigenes were assigned into the three major classification categories: biological process, molecular function, and cellular component (Figure 5). In the biological process category, there were five major representative subgroups with over 10,000 transcripts: cellular process (GO:00099871), metabolic process (GO:0008152), response to stimulus (GO:0050896), biological regulation (GO:0065007), and regulation of biological process (GO:0050789). In the molecular function category, two subgroups, binding (GO:0005488) and catalytic activity (GO:0003824), were predominant. In the cellular component category, only two subgroups were present: cellular anatomical entity (GO:0110165) and protein-containing complex (GO:0032991).

For exploration of biological functions and interactions, *B. koreana* UniTransModels were queried against the Kyoto Encyclopedia of Genes and Genomes database (KEGG). In total, 12,362 transcripts were found in the 151 KEGG pathways, which involved five functional categories (Appendix A, Table 3). Among these pathways, the “metabolism” pathway included the most transcripts, representing 93% of the data sets. In metabolism, “metabolism of cofactors and vitamins”, “carbohydrate metabolism”, “nucleotide metabolism”, and “amino acid metabolism” were the most abundant unigenes (Appendix A; Table 3).

### 2.4. Isoforms in Berberine Biosynthesis

Of the 23,246 unigenes identified, 14,767 (64.21%) did not have isoforms, and the rest produced isoforms numbering from 2 to 31 (Table 2), with an average of 15 isoforms per gene. *O*-methyltransferase and oxidases are major enzyme families in BIA synthesis [5]. In our dataset, the number of unigenes was 46 and 139 for *O*-methyltransferases and CYP oxidases, respectively (Appendix A). Among the 15 genes involved in berberine biosynthesis, the number of paralogues per gene ranged from 1 to 22, such that BBE, STOX, and SOMT had no paralogue, whereas 3OHase had 22 paralogues (Table 1). The number of isoforms ranged from 1 to 10, with an average of 3.6 per gene. Figure 4 shows the structures of isoforms of PB10989, which is one of the 11 paralogues of the NCS gene family. The number of exons in PB10989 ranged from one (PB10989.4) to four (PB10989.3, PB10989.7, PB10989.8, PB10989.9).

Phylogenetic analysis of the NSC families in *B. koreana* and NSC enzymes from other species producing BIAs in Ranunculales and *N. nucifera* in Proteales was performed (Appendix A). For phylogenetic analysis, the longest isoforms were used if more than one isoform was present in *B. koreana*. The eleven isoforms clustered into two major clusters: one was exclusive to *B. koreana* NCSs, and the other cluster included NCSs of other species. The exclusive *B. koreana* cluster included the isoforms PB21974.1, PB7426.1, PB15604.1, PB10989.1, and PB15603.2. The second major cluster was divided into several small groups, with PB.9306.1 being closely placed with *N. nucifera* NCS. PB8681.1, PB8683.1, and PB8682.1 were closest to *Thalictrum thalictroides* and *Sinopodophyllum hexandrum* NCSs. The isoform PB11959.1 tied with others as a solo isoform at a single node.

### 2.5. Phylogenetic Analyses of the Methyltransferases and Oxidase/Reductases among the Berberine Synthesis Enzymes

We identified 46 *O*-methyltransferases and 139 CYP450 oxidase-encoding unigenes in our dataset (Appendix A). Among these, the enzymes involved in the berberine biosynthesis pathway were analyzed for their phylogenetic relationships with homologues in other species in Ranunculales and *N. nucifera* (Appendix A). In the *B. koreana* transcriptome dataset, four *O*-methyltransferases were recognized in the berberine biosynthetic pathway. The numbers of unigenes encoding *O*-methyltransferases were 2, 2, 1, and 4 in 6OMT, 4OMT, SOMT, and CNMT, respectively. Each of the *O*-methyltransferases formed a distinct clade with homologues from other species with high bootstrap values. The CNMT clade was out-formed from the large cluster of 6OMT, 4OMT, and SOMT with low bootstrap values. In oxidases/reductases, the overall tree showed two large clusters: one with CYP oxidases and another with BBE and STOX (Figure 6). In the cluster of CYPs, two subclusters were formed, one with CYP719A21/CA and another with CYP80A, CYP80G, and CYP80B/NMCH. In the CYP80 subclade, the CYP80B/NMCH members separated from those of CYPA and CUP80G. The numbers of unigenes for CYP719A21/CAS, CYP80A, CYP80G, CYP80B/NMCH, BBE, and STOX were 3, 2, 4, 2, 1, and 1, respectively, and each of these enzymes also formed a distinct clade with homologues from other species except for two CYP80A enzymes. The large cluster of BBE and STOX was robust with a bootstrap value of 100.

## 3. Discussion

De novo Iso-seq transcriptome sequencing analysis was carried out with the publicly available bioinformatics tool Cogent in the medicinal plant *B. koreana*. Cogent is specifically designed to use in cases where there is no reference genome available [22,23]. With the lack of a reference genome in *B. koreana*, we built a bioinformatics pipeline using Cogent for de novo genome assembly and annotation of transcriptomes using high-quality Iso-Seq data (Figure 2). Our analysis proved that this pipeline could be applied to characterize full-length transcriptomes in any other species lacking a reference genome.

PacBio Iso-Seq allows reading long transcripts with relatively low error rates because the reads are derived from the results of multiple sequencing of circular cDNA in SMRT cells [18,21]. In our analysis, the average length of the transcripts was 3059 bp, and the longest read was 13,544 bp. Long reads have several advantages, including better chances for Basic Local Alignment Search Tool (BLAST) matching with annotated databases, identification of isoforms, and identification of gene families. In the current study, 86.16% (20,029) and 69.9% (16,179) of the 23,246 identified unigenes matched with the Nt and Nr databases, which are lower than the percentages of matches for ginseng (95.8%) [20]. This suggests that there are still many novel or uncharacterized transcripts in *B. koreana*. Similar results have been reported in another medicinal herb, *Z. planispinum* [21]. The higher matching of the unigenes of *B. koreana* with lotus (*N. nucifera*) than with opium poppy (*P. somniferum*) unigenes was odd because the genera *Berberis* and *Papaver* belong to the order Ranunculales, but the genus *Nelumbo* belongs to the order Proteales. The species in Ranunculales produce a variety of alkaloids [24], and opium poppy produces morphine, codeine, and noscapine, but not berberine [4,5]. SMRT analysis of *C. deltoidea*, a plant of Papaveraceae in Ranunculales, also revealed that 29.01% of the full-length transcripts showed significant matching with those of lotus [7]. Lotus produces over 200 natural compounds, including alkaloids, glycosides, flavonoids, and terpenoids [25], and genes involved in the BIA biosynthetic pathway were thoroughly investigated for their transcriptional regulation by transcriptome analysis. Our GO analysis results corroborated results for the alkaloid-producing medicinal plants *Z. planispinum* [21] and *C. deltoidea* [14], specifically that the majority of the transcripts were in biological processes with cellular processes and metabolic processes being dominant. The KEGG results of *B. koreana* revealed that a high number of transcripts were assigned to the metabolism pathways of vitamins and cofactors, carbohydrates, nucleotides, and amino acids, which is meaningful because various phytochemicals are derived from these molecules. For example, BIAs are derived from an amino acid tyrosine, which is congruent with the results of a transcriptome study of *Z. planispinum* [21].

Gene duplications are common in eukaryotic species, and genes for secondary metabolism are often present in families that are derived from gene duplication [26]. The duplicated genes were diversified into new pathways in synthesizing secondary metabolites, including BIAs [7,27,28]. In lotus, *CYP80* was duplicated into two copies, *NnCYP80A* and *NnCYP80G*, that play important roles in two groups of alkaloids, aporphine-type BIAs and bis-BIAs [7]. He et al. in 2018 [28] reported similar results in *Coptis* species, where BIA synthesis genes were present mostly in families, but the copy numbers varied for some genes between *Coptis teeta* and *Coptis chinensis*. In the current study, thirteen of the sixteen genes encoding berberine synthesis enzymes were present in more than one copy in *B. koreana*. The sizes of family genes were comparable with those of genes in the *Coptis* species [28]. The 3OHase gene had 22 copies in *B. koreana*, whereas it had six to seven copies in *Coptis*. A notable difference was BBE, which was present in only a single copy in *B. koreana*, but eight copies were present in *Coptis* species. BBEs form a subgroup of superfamilies of FAD-linked oxidases that catalyze the conversion of (*S*)-reticuline to (*S*)-scoulerine, which is a common precursor for berberine, jatrorrhizine, coptisine, and other BIAs [29]. The presence of diverse BIAs in *Coptis* species might have resulted in the expansion of the BBE genes. Gene family size variation shapes natural variation for adaptation in various plant species [30]. For instance, the gene encoding the DBOX enzyme, the last catalytic enzyme in the sanguinarine pathway, was expanded to 35 copies in the medicinal plant *Macleaya cordata*, but the number of *GBOX* gene copies was only one in grapevine and ten in *Arabidopsis* [31]. We also demonstrated that the duplicated copies in each gene diversified and adapted to be differentially expressed between tissues or in growth stages, which was evidenced by other genes involved in plant secondary metabolism [26]. In lotus, most of the BIA synthesis genes showed higher expression in leaf tissues of a high-BIA cultivar than in a low-BIA cultivar, also revealing stage-specific expression [7]. Oxidase/reductases and methyltransferases are major catalytic enzymes in BIA biosynthesis. The cytochrome P450s (CYPs) are an enzyme superfamily present in all kingdoms of life [32], and we identified 124 copies of CYP450 oxidases in the *B. koreana* transcripts. Three CYP subfamilies, CYP80, CYP82, and CYP719, were identified in the BIA biosynthetic pathway [5]. CYP80 and CYP719 subfamilies were present, but CYP82 was absent in *B. koreana* transcripts. CYP82 is involved in the synthesis pathway of noscapine and sanguinarine [5], but *B. koreana* does not synthesize these BIAs. Three members of the CYP80 subfamily were identified in the BIA biosynthetic pathway: CYP80G2, CYP80B3, and CYP80A1. We found seven unigenes encoding CYP80, and they were further classified as CYP80A (PB5792.2), CYP80B (PB21419.1 and PB5745.1), and CYP80G (PB6475.1, PB6474.1, PB10724.1, PB10725.1). CYP80B3 is synonymous with (*S*)-*N*-methylcoclaurine 3′-hydroxylase/*N*-methylcoclaurine 3′-monooxydase (NMCH), which catalyzes the C-3′ hydroxylation of (*S*)-*N*-methylcoclaurine to produce (*S*)-3′-hydroxy-*N*-methylcoclurine in the berberine synthesis pathway [27]. CYP80A is a berbamunine synthase that converts (*S*)-*N*-methylcoclaurine to berbamunine [33]. CYP80G is an (*S*)-corytuberine synthase that converts (*S*)-reticuline to corytuberine [34]. In phylogenetic analysis (Figure 6), the CYP80A and CYP80G members clustered with the CAS/CYP719 clade, which catalyzes the conversion of (*S*)-tetrahydrocolumbamine to (*S*)-canadine [35]. There were three unigenes (PB4732.1, PB4733.1, PB6044.1) encoding CAS/CYP719A2 in the transcriptomes of *B. koreana*.

*O*-methyltransferase (OMT) is an enzyme that transfers methyl groups on a molecule. Four subfamilies (6OMT, 4OMT, SOMT, and CNMT) of the *O*-methyltransferases were involved in berberine synthesis [5]. Of the 27 *O*-methyltransferases identified in our dataset, nine belonged to four subfamilies. Specifically, 2, 2, 1, and 4 were in 6OMT, 4OMT, SOMT, and CNMT, respectively (Appendix A). 6OMT is (*S*)-norcoclaurine 6-*O*-methyltransferase, which catalyzes the conversion of (*S*)-norcoclaurine to (*S*)-coclaurine [36]. Two copies of 6OMT (PB21937.1 and PB7731.1) were found in the transcriptomes, and these two genes clustered together in a deep branch with the 6OMT enzymes from *Thalictrum* and *Coptis*. (*S*)-Coclaurine is converted to (*S*)-*N*-methylcoclaurine by (*S*)-coclaurine-*N*-methyltransferase (CNMT) [37]. There were four copies of CNMT unigenes, PB7238.1, PB7239.1, PB6946.1, and PB7290.1, between which PB7238.1 and PB7239.1 had almost identical amino acid sequences (98.05%), suggesting that the duplication of these two copies occurred recently. However, we could not confirm that they were duplicated in tandem due to the absence of a reference genome sequence. (*S*)-*N*-Methylcoclaurine can also be converted to berbamunine by BS/CYP80A1 (berbamunine synthase), which was present in two copies in our transcriptome dataset. 4OMT is 3′-hydroxy-*N*-methyl-(*S*)-coclaurine 4′-*O*-methyltransferase, which catalyzes the conversion of (*S*)-3′-hydroxy-*N*-methylcoclaurine to (*S*)-reticuline [38]. Two copies (PB7613.1, PB7614.1) of the 4OMT of *B. koreana* that were 89.80% similar were linked in a deep branch, which may also indicate recent duplication. SOMT is (*S*)-scoulerine 9-*O*-methyltransferase, which catalyzes the conversion of (*S*)-scoulerine to (*S*)-3′-hydroxy-methylcoclaurine [39]. One SOMT copy (PB7760.1) was found in our dataset, which was tied with the SOMTs of *Thalictrum* and *Coptis*.

One of the key findings of the genomics studies is that the number of protein coding genes is much less than expected. In addition, the number of genes does not vary greatly between developmentally complex organisms and simple organisms, which was proposed as the G-value paradox [40]. Alternative splicing can partially account for the low number of protein-coding genes and the G-value paradox. SMRT sequencing can analyze the isoforms from alternate splicing in plant transcriptomes (An et al. 2018). Approximately 35.8% of the unigenes had isoforms in the transcript dataset in *B. koreana*, which is higher than the number from Iso-Seq analysis (17.6%) of *Z. planispinum* [21]. The average 1.7 isoforms per gene is lower than the average 3.93 isoforms per gene in cotton [41] and 6.56 isoforms per gene in maize [18]. TYDC, 3OHase, NCS, and COR exhibited the highest number of paralogues and isoforms among the genes involved in berberine biosynthesis (Table 1, Appendix A). Plants produce a high number of non-functional isoforms to avoid the hostile effects and metabolic cost that occur under the effects of stress when functional proteins provide resistance to plants [42]. Producing multiple alternative splicing sites and producing complete protein saves time and energy in transcriptional activation and the accumulation of necessary mRNAs [43].

The biosynthetic pathway of BIAs has been well established in the genera *Papaver* [5], *Coptis* [14,44], and *Nelumbo* [7]. However, no report is available for the berberine pathway in *B. koreana*, and our report is the first transcriptome study in the genus *Berberis*. In the current study, the genes encoding enzymes for the pathway of berberine biosynthesis were identified, but the genes for enzymes in the pathways for other BIAs were not present in our dataset except for corytuberine synthase (CTS) and berbamunine synthase (BS) (Figure 1). (*S*)-Norcoclaurine, an immediate precursor for all BIAs, is converted to berberine by eight sequential biochemical reactions in which seven intermediate protoberberine molecules are involved. Of the seven protoberberine molecules, some are precursors for the branching pathway for other BIAs, and these branching pathways are blocked by the absence of the genes encoding appropriate enzymes. For instance, (*S*)-coclaurine is a precursor molecule for either papaverine synthesis or (*S*)-reticuline [45]. However, the pathway leading to papaverine is blocked by the absence of the gene encoding N7OMT in *B. koreana*. (*S*)-reticuline can be converted into either (*S*)-scoulerine by BBE [46] or (*R*)-reticuline by reticuline epimerase (REPI) [5]. (*R*)-reticuline is a precursor for the synthesis of codeine and morphine [5]. In our dataset, BBE was present, but REPI was absent, so codeine and morphine were not synthesized in *B. koreana*. (*S*)-canadine can be converted to either berberine by STOX or (*S*)-N-methylcanadine by TNMT to lead to noscapine synthesis [47]. TNMT was absent in the transcriptome dataset so that this pathway was blocked, and the synthesis of berberine could proceed by STOX in *B. koreana*. Thus, our results clearly demonstrated how *B. koreana* produces berberine by blocking the pathways leading to other BIAs, but effectively allowing the pathway leading to berberine synthesis. In conclusion, the first research about BIA biosynthesis pathway at the transcriptome level in *B. koreana* was presented. It demonstrated distinctive characteristics in difference between *B. koreana* and others such as lotus. We believe this study can be a fundamental resource for further research to understand berberine as a bioactive compound for food and drug applications.

## 4. Materials and Methods

### 4.1. Plant Material and Storage

Five tissue samples (young and mature leaves, flower, young, and mature fruits) of *B. koreana* were obtained from a stand planted at the experimental garden of Hallym University, which was originally collected at Kangwon province of Korea. Collected tissues were immediately frozen with liquid nitrogen and stored at −80 °C until use.

### 4.2. Illumina RNA-Seq Library Construction and Sequencing

The total RNA was extracted using Hybrid-R RNA extraction kit (Geneall Biotechnology, Seoul, Korea) according to the manufacturer’s protocol. For Illumina library construction, extracted RNA from five different tissues with high RNA integration number (RIN) values (>8) were used for cDNA synthesis. The cDNA was sheared with an average of 500 bp fragment sizes. The TruSeq Library Preparation Kit (Illumina Inc., San Diego, CA, USA) was used to construct the DNA library according to the manufacturer’s protocol. The DNA libraries were sequenced with 150-bp paired-end sequencing using an Illumina Hiseq2500. The quality of the constructed libraries was confirmed by a LabChip GX system (PerkinElmer, Waltham, MA, USA).

### 4.3. Differential Gene Expression (DEG) Analysis

Illumina reads were aligned using Bowtie 2 v2.4.2 [48]. Using RSEM (v1.1.12) [49], the read count values were directly obtained and converted to fragments per kilobase of transcript per million mapped reads (FPKM) values. Then, the DEGs between different tissue samples (leaf vs. flower, young leaf vs. mature leaf, and young fruit vs. mature fruit) were detected with the standardization method TMM using edge R [50]. The significant DEGs were screened at false discovery rates (FDRs) <0.05 and absolute log2fold change values >0.01.

### 4.4. Full-Length cDNA Sequencing

The same amount of total RNA from five tissues (flower, young leaf, mature leaf, young fruit, and mature fruit) were pooled and subjected to RNA quality check (Agilent Technologies, Santa Clara, CA, USA), followed by cDNA synthesis for full length cDNA sequencing analysis. The cDNA size selection was performed through a BluePippin (Sage Science, Beverly, MA, USA) to build a cDNA library of size <4 kb and >4 kb cDNA. Iso-Seq library preparation and sequencing were done using PacBio full-length cDNA library and sequencing kit according to the manufacture’s protocol (Pacific Biosciences of California, Inc., Menlo Park, CA, USA) in sequencing service provider, Theragen Bio. (Theragen Bio Inc., Sungnam, Korea).

### 4.5. Iso-Seq Data Processing with a Standard Bioinformatics Pipeline

Raw sequencing data processing was performed by a standard Iso-Seq protocol in SMRTlink4.0 software. Polymerase reads <50 bp were removed, and the obtained subread BAM files were processed into error-corrected circular consensus (CCS) using the following parameters: full passes ≥0 and predicted consensus accuracy >0.75. By identifying the 5′- and 3′-adapters and the poly (A) tail, full-length and non-full-length reads were classified as full-length and non-full-length. CCSs with all 5′- and 3′-reads were referred to as non-full-length reads, whereas those with all three elements that did not contain any additional copies of the adapter sequence within the DNA fragment were referred to as full-length non-concatemer (FLNC) reads. FLNC reads were clustered into consensus sequences using the Iterative Clustering for Error Correction (ICE) algorithm (https://www.pacb.com/products-and-services/analytical-software (accessed on 3 May 2021)). These reads were combined with non-full-length transcripts and were further polished in clusters using Quiver [51] using the parameters hq_quiver_min_accuracy 0.99, bin_by_primer false, 300 bin_size_kb 1, qv_trim_5p 100, and qv_trim_3p 30 to obtain full-length high-quality (HQ; above 99% accuracy) and low-quality (LQ) polished consensus sequences.

### 4.6. Full-Length Unique Transcript Model Reconstruction

Error-corrected HQ and LQ full-length polished consensus transcripts were merged to remove redundancy using the CD-HITv4.6 package with the parameters -c 0.99–G 0–aL 0.00–aS 0.99–AS 30–M 0–d 0–p 1 [52]. The non-redundant transcripts were processed with the Coding GENome reconstruction Tool (Cogent v7.0.0, https://github.com/Magdoll/Cogent (accessed on 3 May 2021)). In general, Cogent first creates the k-mer profile of non-redundant transcripts, calculates pairwise distances, and then clusters transcripts into families based on their k-mer similarity. Each transcript family was further reconstructed into one or several unique transcript model(s) (referred to as UniTransModels) using a De Bruijn graph method.

### 4.7. Isoform Identification

Error-corrected non-redundant transcripts (transcripts before cogent reconstruction) were mapped to UniTransModels using Minimap2 v2.6 [53]. Splicing junctions for transcripts mapped to the same UniTransModels were examined, and transcripts with the same splicing junctions were collapsed using Cupcake ToFU v13.0.0 [21]. Collapsed transcripts with different splicing junctions were identified as transcription isoforms of UniTransModels.

### 4.8. Functional Annotation and Phylogenetic Analysis

To obtain annotation information of unigenes, transcripts were mapped into various databases. We compared non-redundant protein sequences (Nr) and NCBI nonredundant nucleotide sequences (Nt) against the NCBI database by BLAST v2.10.1 with an E-value cut-off of 1e-5. The Kyoto Encyclopedia of Genes and Genomes (KEGG) and Genome Ontology (GO) analyses were carried out by BLAST2GO v5.2.5 with an E-value cut-off of 1e-5. Phylogenetic analysis was performed using the protein sequences of genes involved in berberine biosynthesis. The genes were aligned with the same genes of closely related families, and a maximum likelihood phylogenetic tree was built using MEGA X (v.10.2.4) [54].

## Figures and Tables

**Figure 1 plants-10-01314-f001:**
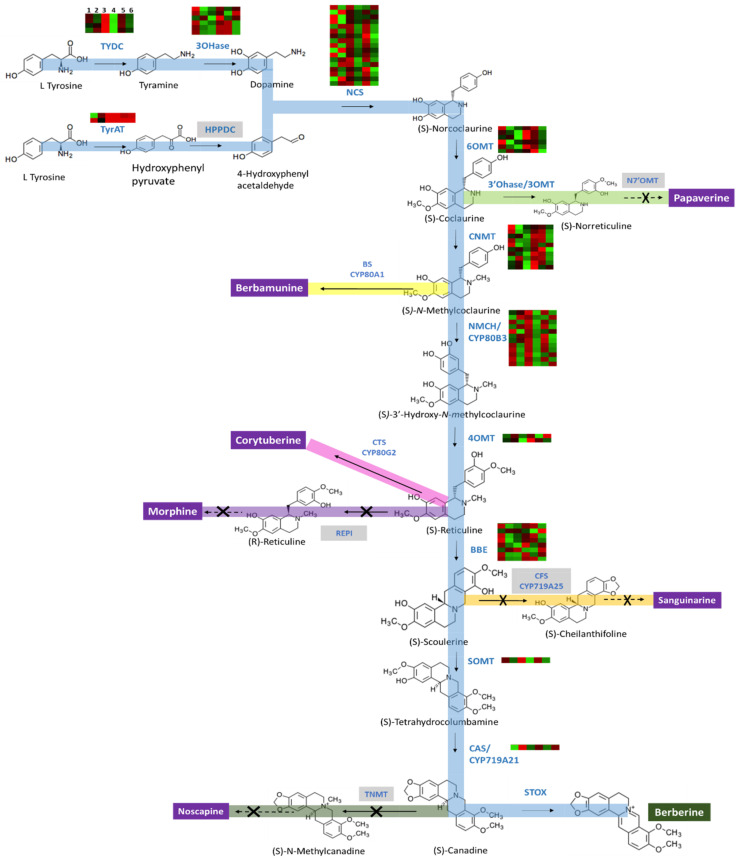
Berberine biosynthesis pathway in *B. koreana*. Enzymes expression patterns are indicated at the side of each step with the value of log fold change. The expression pattern of each unigene is shown within six column grids, with the left to right in the following order, (1) leaf, (2) flower, (3) young leaf, (4) mature leaf, (5) young fruit, and (6) mature fruit. Genes for enzymes in the grey boxes were not found in B. koreana. Solid arrows indicates single step reaction and dashed arrows indicate multiple step reaction. Crossed arrow represents the reaction is terminated in that step. Acronyms: TYDC (tyrosine decarboxylase); 3OHase tyrosine/tyramine 3-hydroxylase/tyrosine 3-monooxygenase; NCS ((*S*)-norcoclaurine synthase); 6OMT ((*S*)-norcoclaurine 6-*O*-methyltransferase); CNMT ((*S*)-coclaurine *N*-methyltransferase); NMCH ((*S*)-*N*-methylcoclaurine 3′-hydroxylase, CYP82B subfamily); 4′OMT ((*S*)3′-hydroxy *N*-methylcoclaurine 4′-*O*-methyltransferase); BBE berberine bridge enzyme (reticuline oxidase); CAS/Cyp719A21 (*S*)-canadine synthase; STOX (*S*)-tetrahydroprotoberberine oxidase; N7ÓMT (norreticuline-7-*O*-methyltransferase); TNMT (tetrahydroprotoberberine*N*-methyltransferase); CFS/CYP719A25 (cheilanthifolinesynthase); REPI (1,2-dehydroreticuline synthase/reductase) CTS (corytuberine synthase); and BS/CYP80G2 (berbamunine synthase).

**Figure 2 plants-10-01314-f002:**
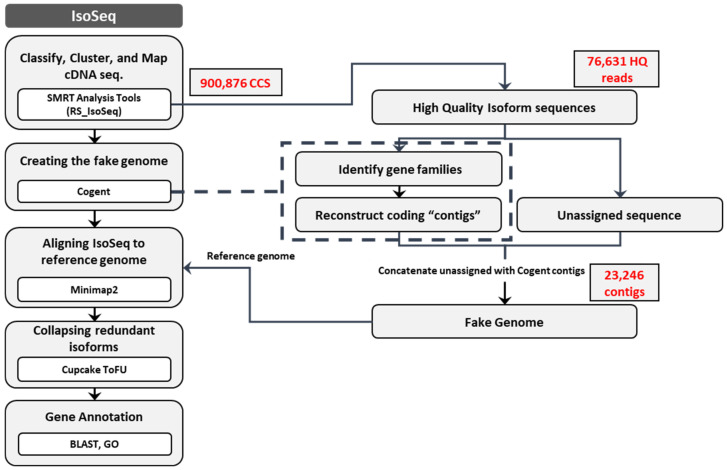
Schematic representation of pipeline to isoforms of full-length complementary DNA (cDNA) sequence used for *B. koreana*.

**Figure 3 plants-10-01314-f003:**
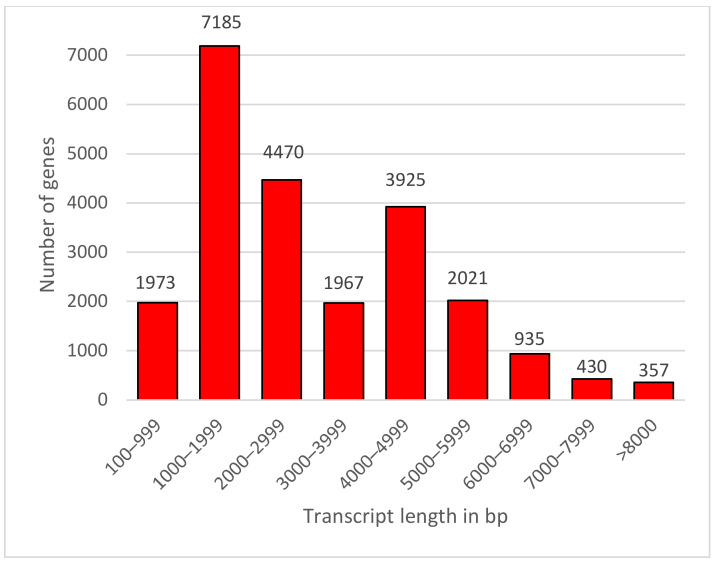
Transcripts length distribution of de novo assemblies in *B. koreana*.

**Figure 4 plants-10-01314-f004:**
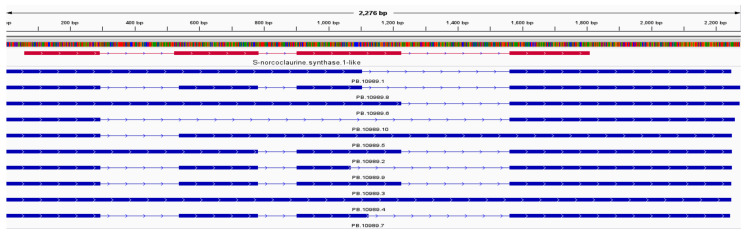
(*S*)-Norcoclaurine synthase (NCS) transcriptome exhibiting isoforms; the red track represents the coding regions (CDSs) of the reference sequence according to the BLAST results, and the blue represents the exons of each isoform.

**Figure 5 plants-10-01314-f005:**
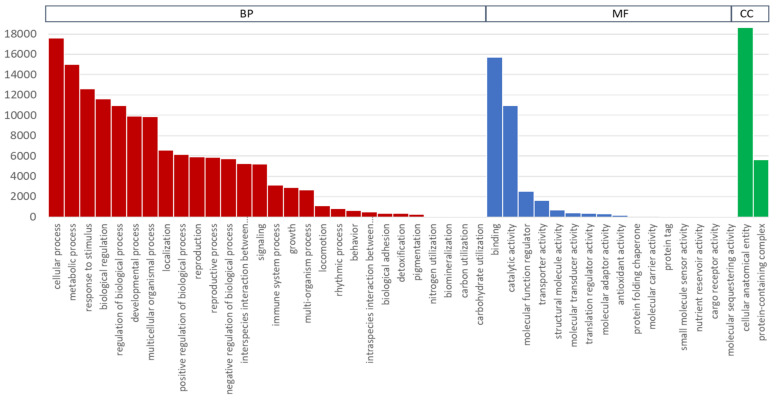
Gene ontology function classifications of unigenes into three main aspects; biological process (BP), molecular function (MF), and cellular component (CP).

**Figure 6 plants-10-01314-f006:**
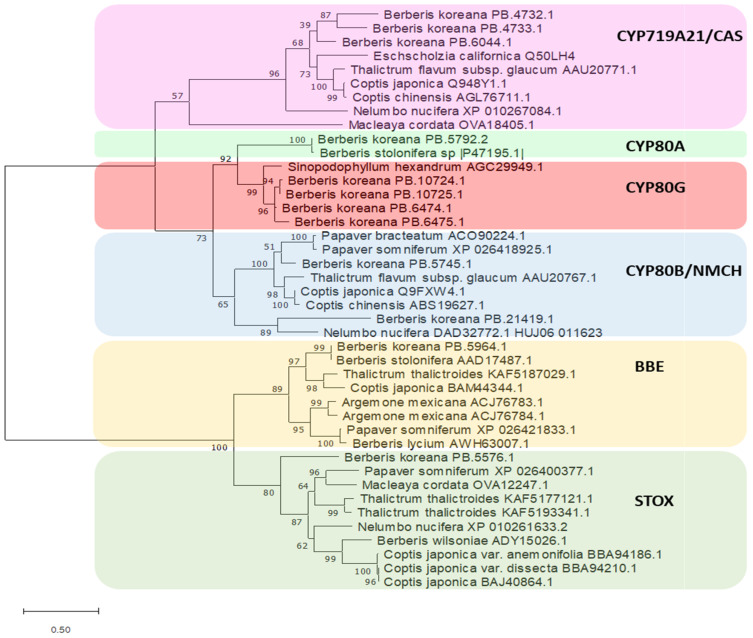
Phylogenetic tree generated using maximum likelihood for CYPs and oxidases based on the deduced amino acid sequences for the *B. koreana* sequences. GenBank accession numbers for the sequences used are as follows: *Sinopodophyllum hexandrum* AGC29949.1, *Papaver bracteatum* ACO90224.1, *Papaver somniferum* XP_026418925.1, *Thalictrum flavum* subsp. *glaucum* AAU20767.1, *Coptis japonica* Q9FXW4.1, *Nelumbo nucifera* DAD32772.1 HUJ06_011623, *Coptis chinensis* ABS19627.1, *Nelumbo nucifera* XP_010261633.2, *Papaver somniferum* XP_026400377.1, *Thalictrum thalictroides* KAF5177121.1, *Macleaya cordata* OVA12247.1, *Coptis japonica* var. *anemonifolia* BBA94186.1, *Thalictrum thalictroides* KAF5193341.1, *Coptis japonica* var. *dissecta* BBA94210.1, *Coptis japonica* BAJ40864.1, *Berberis wilsoniae* ADY15026.1, *Thalictrum flavum* subsp. *glaucum* AAU20771.1, *Coptis japonica* Q948Y1.1, *Coptis chinensis* AGL76711.1, *Eschscholzia californica* Q50LH4, *Nelumbo nucifera* XP_010267084.1, *Macleaya cordata* OVA18405.1, *Berberis stolonifera* sp. |P47195.1|, *Berberis stolonifera* AAD17487.1, *Thalictrum thalictroides* KAF5187029.1, *Coptis japonica* BAM44344.1, *Argemone mexicana* ACJ76783.1, *Papaver somniferum* XP_026421833.1, *Argemone mexicana* ACJ76784.1, *Berberis lycium* AWH63007.1.

**Table 1 plants-10-01314-t001:** Number of paralogues and isoforms of the enzymes involved in benzylisoquinoline alkaloid biosynthetic pathway in *B. koreana*.

Enzymes	Abbreviation	EC Numbers	No. of Paralogues	Range of Isoforms
Tyrosine aminotransferase	TyrAT	2.6.1.5	5	1–2
Tyrosine decarboxylase	TYDC	4.1.1.25	11	1–3
Tyrosine/tyramine 3-hydroxylase	3OHase	1.14.16.2	22	1–4
(S)-norcoclaurine synthase	NCS	4.2.1.78	11	1–10
(RS)-norcoclaurine 6-*O*-methyltransferase	6OMT	2.1.1.128	2	1
(S)-coclaurine *N*-methyltransferase	CNMT	2.1.1.140	5	1–5
(S)-*N*-methylcoclaurine 3′-hydroxylase/*N*-methylcoclaurine 3′-monooxygenase	NMCH/CYP80B3	1.14.13.71	7	1–3
3′-hydroxy-*N*-methyl-(*S*)-coclaurine 4′-*O*-methyltransferase	4′OMT	2.1.1.116	2	1
Berberine bridge enzyme (reticuline oxidase)	BBE	1.21.3.3	1	1
(*S*)-tetrahydroprotoberberine oxidase	STOX	1.3.3.8	1	1
(*S*)-scoulerine 9-*O*-methyltransferase	SOMT	2.1.1.117	1	1
Canadine synthase enzyme	CAS/CYP719A21	1.14.21.5	3	1–2
Codeinone reductase	COR	1.1.1.247	12	1–7
Salutaridine reductase	SalR	1.1.1.248	6	1–5
Codeine *O*-demethylase	CODM	1.14.11.32	6	1–4
3′-*O*-methyltransferase	3OMT	2.1.1.267	4	1–5

**Table 2 plants-10-01314-t002:** Isoseq result and isoform number.

**Iso Seq Result**	**Number of Reads**	**Length (bp)**
High quality consensus Seq.	76,631	216,086,311
Reconstructed Coding Contig	19,902	60,494,776
Unassigned Seq	3344	10,608,597
Fake Genome	23,246	71,103,373
Minimun read length		100
Maximum read length		13,544
Average read length		3059
**Number of Isoforms**	**Number of Transcripts**	**Percentage (%)**
1	14,767	64.21
2	4812	20.92
3	1680	7.30
4	728	3.17
5	393	1.71
6	209	0.91
7	132	0.57
8>	525	2.27

**Table 3 plants-10-01314-t003:** Number of enzymes and unigenes in *B. koreana* based on KEGG classification.

KEGG Category	KEGG Subcategory	Number of Pathways	Number of Enzymes	Number of Unigenes	Percentage
Metabolism	Biosynthesis of other secondary metabolites	23	140	574	4.64
	Amino acid metabolism	14	314	1189	9.62
	Galactose metabolism	1	9	9	0.07
	Metabolism of terpenoids and polyketides	11	68	235	1.90
	Metabolism of other amino acids	8	74	328	2.65
	Nucleotide metabolism	2	81	1765	14.28
	Lipid metabolism	16	185	1019	8.24
	Metabolism of cofactors and vitamins	13	153	2322	18.78
	Carbohydrate metabolism	15	358	1987	16.07
	Energy metabolism	7	99	691	5.59
	Xenobiotics biodegradation and metabolism	15	70	965	7.81
	Glycan biosynthesis and metabolism	15	81	376	3.04
Human Diseases	Infectious disease: viral	3	3	3	0.02
	Cancer: overview	1	1	240	1.94
	Drug resistance: antimicrobial	1	1	1	0.01
Environmental information processing	Signal transduction	3	25	142	1.15
Genetic information processing	Translation	1	21	21	0.17
Organismal System	Immune system	2	3	495	4.00

## Data Availability

All data were submitted to the National Centre for Biotechnology Information (NCBI) under SRA number PRJNA705056.

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
