# Peer review of "Gene Expression and Isoform Identification of PacBio Full-Length cDNA Sequences for Berberine Biosynthesis in Berberis koreana"

_plants, 2021, doi:10.3390/plants10071314_

Round 1

Reviewer 1 Report

This is a well-written and organised account of a transcriptome analysis of Berberis koreana. It adds to information about benzylisoquinoline alkaloid biosynthesis and describes use of several complementary sequencing technologies. It provides useful information and merits publication. The manuscript needs minor proofreading for some typos and formats to conform to the journal's style. The Results, section 2.1, paragraph 2, starts with 'As expected'. Please could be authors explain why these results are expected. It is not obvious from the Introduction. In the third paragraph (starts The genes for TYDC, NCS, SOMT, ...) please can the authors cite the Table/Figure where these data for transcripts in particular tissues are shown.

Author Response

Response – Dear Reviewer,

Thank you for your detailed comments. We agree with your comment, however, in the introduction Page 2 Line 3, it is explained that berberine is the main BIA found in Berberis species and thus we will only expect to see the unigenes for genes encoding enzymes that are in the Berberis biosynthesis pathway. This is also elaborated with examples of other BIAs that were not found in B koreana. Thus, we did not change the sentence, “As expected”

For your second comment on the third paragraph, we have added (Fig 1) on the given sentence (Page 4 line3)

Reviewer 2 Report

The publication proposal has been written in a clear and transparent manner that does not raise any doubts. The first part introduces the reader to the problem well. Methodically, the study was very well prepared and well described. Unfortunately, the authors did not avoid mistakes, there is a few egzamples: no italics in the name "B. koreana ”in the summary and the last paragraph of the introduction; no dots in brackets (B koreana) in the first paragraph of the introduction and in the description of figure 2 as well as in the name of N nucifera (point 2.4); in many places in the text the authors use the phrase "Fig. X ", in the remaining "Figure X"; in the description of figure 3 should be "B. koreana"; a lot of unnecessary commas, etc.

However, these are minor editorial errors that do not affect the factual reception of the text. However, I recommend that you carefully check the text and introduce editorial corrections.

The cDNA library should be available in one of the available databases, and detailed data describing the results obtained, such as functional analysis or expression patterns of individual genes, should be attached to the supplementary material.

Author Response

Dear Reviewer,

Thank you for the time and effort you have put into the comments. We have made all the necessary correction not only in the mentioned examples but also throughout the manuscript. All changes are highlighted in red.

In addition, RNASeq data has been uploaded and the SRA number is provided under section availability of data and material (page 15 Line 12)

Reviewer 3 Report

The manuscript plants-1227368 is devoted to transcriptome study in the Berberis koreana. In this study the genes encoding enzymes for the pathway of berberine biosynthesis were identified, and proper conclusions were made.

The manuscript is clearly and understandably written, it may be of interest to a certain circle of readers, but contains minor flaws They are set out below:

  • There are some abbreviations that are either not deciphered (KEGG, BLAST) or deciphered not at the first mention in the text (NCS). Check please through the text.
  • in the first paragraph on page 2, insert the link to fig. 1
  • in the caption to Fig. 5 decrypt BP, MF and CC

Author Response

Dear Reviewer,

Thank you for your time and effort in the comments. We have added the description of abbreviations as pointed out and made sure it is done throughout the manuscript. The fig 1 link on Page 2 was not added because Fig 1 is a modified pathway, which is modified from the one Hagel et al 2015 and that, is added in the results (Page 4 line 3). However, we have added reference in the mentioned paragraph (Page 2 Line 26).

Fig 5 decryption of BP, MF and CC were made. They are Biological Process, Molecular Function and Cellular Component.